# Physical therapy in the intensive care unit: A cross-sectional study of three Asian countries

**Mary Audrey Domingo Viloria[1,2]◉, Shin-Da Lee[2]◉, Tetsuya Takahashi[ID][3]\*, Yu-Jung Cheng[ID][2]\***

**1** Department of Physical Therapy, College of Health Sciences, Mariano Marcos State University, Batac City, Ilocos Norte, Philippines, **2** Department of Physical Therapy, Graduate Institute of Rehabilitation Science, China Medical University, Taichung City, Taiwan, **3** Department of Physiotherapy, Faculty of Health and Medical Sciences, Juntendo University, Tokyo, Japan

◉ These authors contributed equally to this work.
\* chengyu@mail.cmu.edu.tw (YJC); te-takahashi@juntendo.ac.jp (TT)

## Abstract

### Background

Physical therapy (PT) is beneficial for critically ill patients, but the extent of its application in the intensive care unit (ICU) differs between countries. Here, we compared the extent of PT intervention in the ICU in Japan, the Philippines, and Taiwan by evaluating the sociodemographic and ICU-related profiles of ICU physical therapists.

### Materials and methods

In this cross-sectional study, a semistructured nationwide online survey was distributed to ICU physical therapists in the three countries.

### Results

We analyzed the responses of 164 physical therapists from Japan, Philippines, and Taiwan. Significant differences were observed between the countries in all sociodemographic variables and the following ICU-related profiles of physical therapists: ICU work experience, duration of the ICU posting, number of hours per day spent in the ICU, on-call ICU PT service engagement, source of ICU patient referral, therapist–patient ratio, and ICU-related PT training participation ($p < 0.05$). Medical, surgical, and neurologic ICUs were the most common ICU workplaces of the ICU physical therapists, but only surgical and neurologic ICUs exhibited significant differences between the countries ($p < 0.05$). Standard PT techniques in the ICU were passive and active-assisted range of motion, positioning, and breathing exercises but were implemented with significantly different frequencies between the countries ($p < 0.05$). The most common challenge faced in ICU PT service delivery by respondents from all three countries was lack of training prior to ICU duty, and lack of training was even bigger challenge in Japan than in other two countries after adjustment of age, highest educational attainment, and work experience.

**Data Availability Statement:** All relevant data are within the paper and its Supporting information files.

**Funding:** This work was supported by the Ministry of Science and Technology in Taiwan (111-2410-H-039-004-; NSTC 112-2410-H-039-008), China

Medical University Hospital (DMR-110-172), and China Medical University (CMU111-MF-89). The funders had no role in study design, data collection and analysis, decision to publish, or manuscript preparation. These fundings covered the IRB application fee, publication fee, ethic training courses, and English editing.

**Competing interests:** The authors have declared that no competing interests exist.

## Conclusion

The differences in the health-care system between Japan, the Philippines, and Taiwan were related to differences in the compliance with internationally recommended PT practice standards in the ICU, differences in the type of PT intervention prioritized, and the challenges encountered in ICU PT service delivery.

## Introduction

Critically ill patients with life-threatening conditions, such as organ failure, need to be admitted to the intensive care unit (ICU) and require regular monitoring and other specialized treatments [1]. Advancements in treatment strategies provided at the ICU to critically ill patients have decreased their mortality rate; however, some of the techniques, such as sedation and use of tubes and the attachment of equipment, have also prolonged ICU stay [2, 3]. These patients also require constant bed rest, which results in bed rest–related physiological changes [4–6], including impaired neuromuscular [7] and psychological health. This, in turn, affects physical function [8], manifesting as weakness [9], exercise intolerance, fatigability, difficulty in returning to the usual activities of daily living, and organ system complications [10]. The health-related quality of life (HRQoL) of ICU patients has been reported to be significantly lower than that of the general (healthy) population, even weeks to years after discharge from the ICU [8, 11–14].

Physical therapy (PT) is essential for the management of critically ill patients in the ICU [15, 16]. It involves different techniques conducted by physical therapists [17], such as range of motion and stretching exercises, mobility, functional training, and the use of physical agents or electrotherapy, which help improve the patients' physical function that has been compromised by physiological, pathological, or environmental factors [18]. With improvement of the patients' physical function, their HRQoL also improves [16].

Because critical illness results in ICU admission and both critical illness and ICU admission (or stay) impair physical health and eventually affect HRQoL, various PT techniques can reverse these ICU-acquired complications. Efficient PT administration to ICU patients with acute respiratory failure who are on mechanical ventilation resulted in better physical and mental functional outcomes at hospital discharge than those who received only usual care [19–21]. Among ICU patients on prolonged mechanical ventilation ($\geq$14 days), PT strategies, such as upper and lower extremity exercises, functional retraining exercises, and breathing exercises, were associated with enhanced functional outcomes compared with standard therapy alone [22].

Accumulating evidence worldwide has indicated that PT is beneficial for ICU patients. However, the extent of PT intervention in the ICU varies between countries, regions, or centers (or hospitals). These differences may be due to ICU-related aspects, such as treatment protocols, and ICU-specific challenges encountered by the physical therapist. In some countries such as Sri Lanka [23] and the United States [24], ICU physical therapists implement usual PT management for ICU patients, whereas ICU physical therapists in Nepal appear to focus on chest PT [25]. Physical therapist–related concerns include staffing and training, which influence the extent of PT intervention in the ICU [24, 26]. For example, Jordanian physical therapists do not require any specific ICU-related training [27], whereas US ICU physical therapists have competency requirements for working in an ICU of a teaching hospital, although the training is usually hospital-based rather than a formal training program [24]. Other reported challenges in ICU PT service delivery include patient-related issues, such as prioritization and

perception of the importance of PT [24]; staff-related concerns, such as poor interdisciplinary communication [26, 28]; and system-related problems, such as the lack of consultation criteria [24], and lack of PT service coverage [23].

The variations in PT intervention in the ICU in different countries may be attributed to the country's health-care system. Japan, the Philippines, and Taiwan have different health-care systems. Although numerous studies have reported on the current status of PT intervention in the ICU, no study has compared it between different Asian countries. In the current study, we compared the extent of PT intervention in the ICU between Japan, the Philippines, and Taiwan by thoroughly evaluating the sociodemographic and ICU-related profiles of the physical therapists from the three countries. We hypothesized that given the differences in the countries' health-care systems, the implementation of PT intervention in the ICU would also significantly differ between the countries.

## Materials and methods

### Study design

This cross-sectional study was conducted from November 2021 to February 2022 in Japan, the Philippines, and Taiwan. This study was approved by the Ethics Review Board of China Medical University Hospital (CMUH Protocol Number: CMUH110-REC1-192) and Faculty of Health Sciences Research Ethics Committee of Juntendo University (Execution Permit Number: 00013).

### Participants

We used purposive expert sampling to enroll physical therapists from Japan, the Philippines, and Taiwan with the following inclusion criteria: (1) having worked or currently working in the ICU regardless of age, sex, nationality/race, and years of experience as physical therapists and (2) having an active e-mail. Informed consent was obtained from all respondents; the informed consent from was provided in the first part of the online survey prior to the actual survey questions. The informed consent form contained an explanation of the study purpose and procedures and stated that all data obtained will be treated with utmost confidentiality.

### Study instrument

We used a two-part, semistructured, open online survey which based on Equator Network's Checklist for Reporting Results of Internet E-Surveys (CHERRIES) guidelines [29]. Moreover, the survey was duly validated by two experienced ICU physical therapists. The survey was semipatterned from the study of Sigera et al. (2016), which investigated the PT practice profile of ICU physical therapists in Sri Lanka [23]. The survey included a combination of dichotomous, multiple-choice, and closed-ended questions, which were formulated to be as wideranging as possible, including other PT interventions in the ICU-related aspects deemed appropriate in Japan, the Philippines, and Taiwan. Part I of the survey comprised seven questions regarding the sociodemographic profile of the respondents. Part II included 14 questions regarding the respondents' ICU-related profile, such as ICU work experience, most commonly implemented PT techniques in the ICU, and challenges experienced in ICU PT service delivery. Moreover, the survey included adaptive questions for some items. The English version of the survey was made and translated into Japanese and Chinese by authors for the respondents' easier understanding. An electronic survey, consisting of eight sections, was made using Google Forms. The survey link was posted on the social media (such as Facebook and Instagram) of PT interest groups in Taiwan and the Philippines and of the Japanese Society of Education

for Physicians and Trainees in Intensive Care (JSEPTIC) in Japan. The final survey was evaluated with the Strengthening the Reporting of Observational Studies in Epidemiology (STROBE) statement [30] (S1 Checklist).

## Statistical analyses

The data were analyzed using IBM SPSS Statistics for Windows, Version 22.0. (Armonk, NY: IBM Corp) and SAS 9.4 (SAS Institute Inc., Cary, NC, USA). The respondents' sociodemographic and ICU-related characteristics are presented as frequency (*f*), percentage (%), and mean. For the categorical variables, chi-square or Fisher's exact test were used, and Fisher's exact test was conducted instead of chi-square test if any of the expected values in a cell is less than five. The one-way analysis of variance (ANOVA) was used for continuous variables. The level of significance was set at $p < 0.05$. Although the sample size was not determined, sample size between 30 and 500 for comparative analysis was considered appropriate [31].

## Results

### Sociodemographic profile

Overall, 164 eligible respondents (76, 45, and 43 from Japan, the Philippines, and Taiwan, respectively) were included (Fig 1), and Table 1 summarizes their sociodemographic profiles. All submitted data were included in the analysis.

### ICU-related profile

Considerable variations were noted between Japan, the Philippines, and Taiwan in the different ICU-related factors (Table 2).

Nearly half of the Japanese respondents had ≥10 years of ICU work experience, which was significantly more than that of the respondents from the Philippines and Taiwan ($p < 0.05$). Moreover, most of the respondents in all the three countries were working in medical and surgical ICUs, but significant differences were observed between the countries only in surgical ICU as well in pediatric, neurologic, and cardiac ICUs (all $p < 0.05$). Significant differences were also noted in the duration of the ICU posting, number of hours per day spent in the ICU, engagement in on-call ICU PT services, time of on-call ICU PT, source of ICU patient referral, and physical therapist-to-patient ratio (all $p < 0.05$). None of the other ICU-related variables were significantly different (all $p > 0.05$).

### ICU-related PT training

The engagement of the PT respondents in ICU-related PT training was significantly different between Japan, the Philippines, and Taiwan ($p < 0.05$, Table 3).

Only national training and ICU department–facilitated trainings were significantly different between Japan, the Philippines, and Taiwan ($p < 0.05$). Notably, the respondents from Japan were more engaged in these trainings than those from the Philippines and Taiwan. No significant differences were noted in the mode or duration of training ($p > 0.05$).

The following PT interventions were uncommon in the three countries and were not significantly different between them: forced expiration or huffing technique, active cycle of breathing technique, autogenic drainage, suction technique, manual hyperinflation, cognitive task technique, oral or feeding intervention, and Volta therapy (all $p > 0.05$).

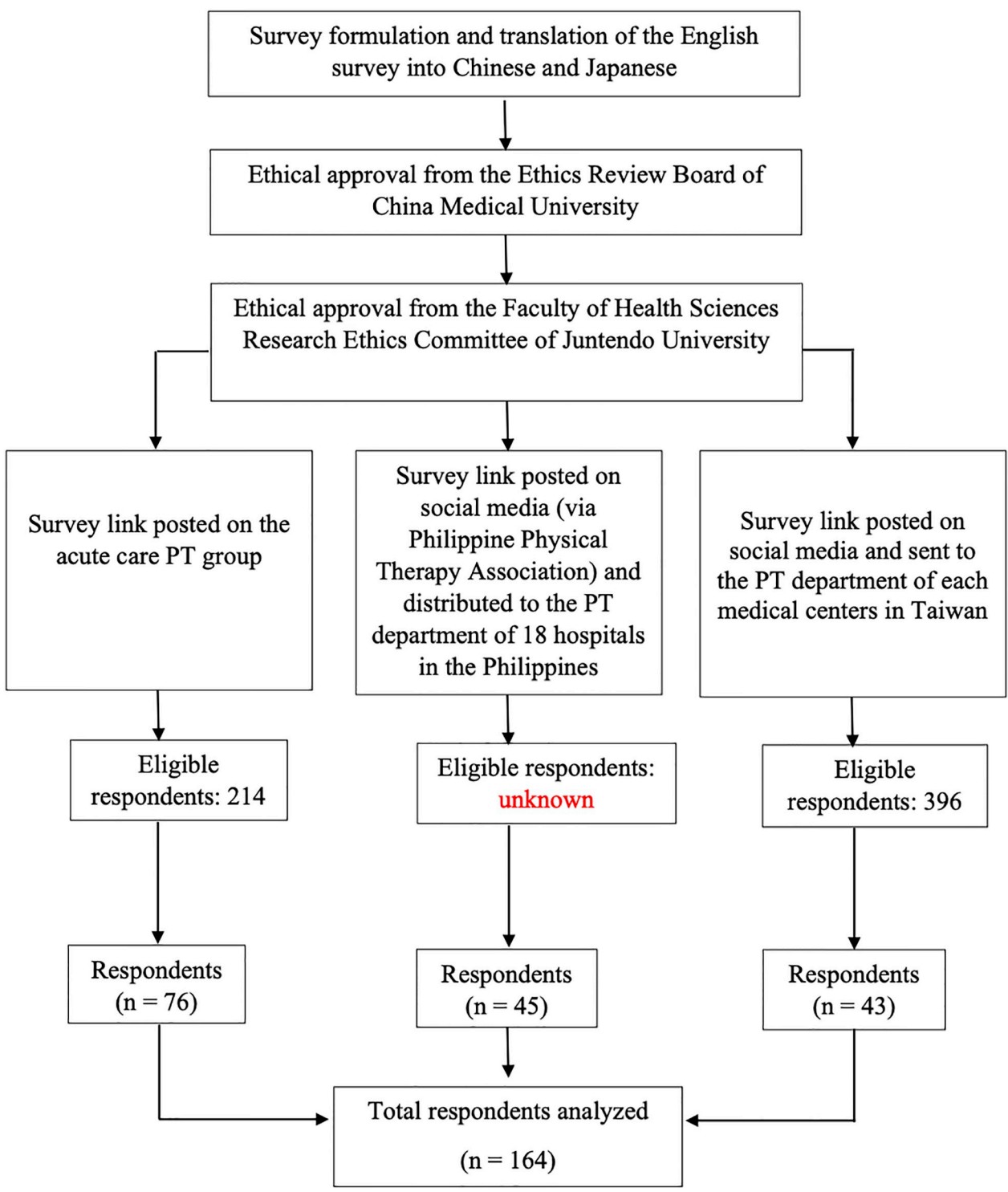

**Fig 1. Study flowchart.** PT, physical therapy.

**Table 1. Sociodemographic profile of respondents.**

| | JP (n = 76) | PH (n = 45) | TW (n = 43) | *p value* |
|---|---|---|---|---|
| **Gender, *n* (%)** | | | | |
| Male | 68 (89.47) | 24 (53.33) | 13 (30.23) | **0.000**[a] |
| Female | 8 (10.53) | 21 (46.67) | 30 (69.77) | |
| **Age (years), *n* (%)** | | | | |
| 21–25 | 3 (3.95) | 14 (31.11) | 5 (11.63) | **0.000**[a] |
| 26–30 | 9 (11.84) | 17 (37.78) | 10 (23.26) | |
| 31–35 | 19 (25) | 7 (15.56) | 7 (16.28) | |
| 36–40 | 17 (22.37) | 4 (8.89) | 7 (16.28) | |
| 41–45 | 13 (17.11) | 0 (0) | 7 (16.28) | |
| >45 | 15 (19.74) | 3 (6.67) | 7 (16.28) | |
| **Highest Educational Attainment, *n* (%)** | | | | |
| Diploma | 23 (30.26) | - | - | **0.000**[b] |
| Bachelor | 30 (39.47) | 43 (95.56) | 30 (69.77) | |
| Master | 8 (10.53) | 0 (0) | 10 (23.26) | |
| Doctorate | 15 (19.74) | 2 (4.44) | 3 (6.98) | |
| **Work Experience, *n* (%)** | | | | |
| < 1 year | 0 (0) | 2 (4.44) | 3 (6.98) | **0.000**[a] |
| 1–5 years | 5 (6.58) | 25 (55.56) | 10 (23–26) | |
| 6–10 years | 18 (23.68) | 12 (26.67) | 9 (20.93) | |
| 11–15 years | 18 (23.68) | 5 (11.11) | 9 (20.93) | |
| 16–20 years | 17 (23.37) | 1 (2.22) | 5 (11.63) | |
| 20 years and more | 18 (23.68) | 0 (0) | 7 (16.28) | |
| **Type of Hospital, *n* (%)** | | | | |
| **1. Level** | | | | |
| Medical/National | 41 (53.95) | 33 (73.33) | 26 (60.47) | **0.000**[b] |
| Regional | 35 (46.05) | 3 (6.67) | 15 (34.88) | |
| Local | 0 (0) | 9 (20) | 2 (4.65) | |
| **2. Ownership** | | | | |
| Government | 28 (26.84) | 28 (62.22) | 12 (27.91) | **0.003**[a] |
| Private | 48 (63.16) | 17 (37.78) | 31 (72.09) | |
| **3. Teaching Affiliation** | | | | |
| Teaching | 51 (67.11) | 27 (60) | 43 (100) | **0.000**[a] |
| Non-Teaching | 25 (32.89) | 18 (40) | 0 (0) | |

JP, Japan; PH, the Philippines; TW, Taiwan; -, not included in the questionnaire; numbers in bold indicate *p* < 0.05

[a] Pearson chi-square test

[b] Fisher's exact test

## Most common PT intervention implemented in the ICU

The Pearson chi-square test (Fisher's exact test) revealed significant differences in the extent of implementation of different PT techniques (Fig 2 and S1 Table).

Our data indicated that postural drainage, sitting balance and tolerance exercise, standing balance and tolerance exercise, and ambulation training were significantly different between the countries, with respondents from Japan implementing these PT interventions more than those from the Philippines and Taiwan (all *p* < 0.05). Moreover, positioning, breathing exercise, passive range of motion, active range of motion, stretching exercise, electrotherapy

**Table 2. ICU-related profile of respondents.**

| | JP (n = 76) | PH (n = 45) | TW (n = 43) | p value |
|---|---|---|---|---|
| **ICU work experience, *n* (%)** | | | | |
| <1 year to 3 years | 12 (15.79) | 25 (57.78) | 24 (55.81) | **0.000**[b] |
| 4–6 years | 20 (26.32) | 11 (24.44) | 5 (11.63) | |
| 7–9 years | 11 (14.47) | 5 (11.11) | 1 (2.33) | |
| 10 years and more | 33 (43.42) | 3 (6.67) | 13 (30.23) | |
| **Type of ICU, *n* (%)** [#] | | | | |
| Medical | 69 (90.79) | 41 (91.11) | 34 (79.07) | 0.125[a] |
| Surgical | 68 (89.47) | 32 (71.11) | 36 (83.72) | **0.034**[a] |
| Neonatal | 16 (21.05) | 9 (20) | 9 (20.93) | 0.990[a] |
| Pediatric | 8 (10.53) | 21 (46.67) | 11 (25.58) | **0.000**[a] |
| Neurologic | 43 (56.58) | 35 (77.78) | 23 (53.49) | **0.030**[a] |
| Cardiac | 65 (85.53) | 13 (28.89) | 31 (72.09) | **0.000**[a] |
| Others | 4 (5.26) | 5 (11.11) | 7 (16.28) | 0.153[b] |
| **Hiring department, *n* (%)** | | | | |
| Rehabilitation/PT | 73 (96.05) | 45 (100) | 40 (93.02) | 0.207[b] |
| ICU | 0 (0) | 0 (0) | 0 (0) | |
| Others | 3 (3.95) | 0 (0) | 3 (6.98) | |
| **Status of ICU posting, *n* (%)** | | | | |
| Full-time | 5 (6.58) | 2 (4.44) | 2 (4.65) | 1.000[b] |
| Temporary | 71 (93.42) | 43 (95.56) | 41 (95.35) | |
| **Duration of posting, *n* (%)** | | | | |
| <1 month | 3 (3.95) | 15 (33.33) | 3 (6.98) | **0.000**[a] |
| 1–3 months | 2 (2.63) | 9 (20) | 0 (0) | |
| Half year | 2 (2.63) | 2 (4.44) | 16 (37.21) | |
| Whole year | 12 (15.79) | 11 (24.44) | 2 (4.65) | |
| Will not rotate | 57 (75) | 8 (17.78) | 22 (51.16) | |
| **ICU stay/day, *n* (%)** | | | | |
| <30–60 minutes | 20 (26.32) | 32 (71.11) | 25 (58.14) | **0.000**[b] |
| 1–2 hours | 18 (23.68) | 12 (26.67) | 10 (23.26) | |
| 3–4 hours | 18 (23.68) | 1 (2.22) | 4 (9.30) | |
| 5–6 hours | 8 (10.53) | 0 (0) | 3 (6.98) | |
| 7 hours and more | 12 (15.79) | 0 (0) | 1 (2.33) | |
| **Engagement in on-call ICU PT service, *n* (%)** | | | | |
| Yes | 27 (35.53) | 20 (44.44) | 0 (0) | **0.000**[a] |
| No | 49 (64.47) | 25 (55.56) | 43 (100) | |
| **Time of on-call ICU PT, *n* (%)** [$] | | | | |
| Within the 6–8 hours work | 23 (85.19) | 8 (40) | not applicable | **0.000**[b] |
| After the 6–8 hours work | 1 (3.70) | 10 (50) | not applicable | |
| During and after the 6–8 hours work | 3 (11.11) | 2 (10) | not applicable | |
| **ICU physical therapists in hospital, (%)** | 36.12 | 73.90 | 31.13 | 0.133* |
| **Source of ICU patient referral, *n* (%)** | | | | |
| Direct access | 0 (0) | 0 (0) | 0 (0) | **0.000**[a] |
| Intensivist | 60 (78.95) | 4 (8.89) | 2 (4.65) | |
| Physiatrist | 1 (1.32) | 22 (48.89) | 3 (6.98) | |
| Intensivist + Physiatrist | 15 (19.74) | 19 (42.22) | 38 (88.37) | |
| **Number of ICU patients/day, *n* (%)** | | | | |

*(Continued)*

**Table 2.** (Continued)

| | JP (n = 76) | PH (n = 45) | TW (n = 43) | *p value* |
|---|---|---|---|---|
| 1–2 | 26 (34.21) | 13 (24.07) | 15 (34.88) | 0.117[b] |
| 3–4 | 18 (23.68) | 15 (27.78) | 6 (13.95) | |
| 5–6 | 13 (17.11) | 12 (22.22) | 7 (16.28) | |
| 7–8 | 6 (7.89) | 4 (7.41) | 7 (16.28) | |
| 9–10 | 5 (6.58) | 0 (0) | 1 (2.33) | |
| >10 | 8 (10.53) | 1 (1.85) | 7 (16.28) | |
| **PT-patient ratio, *n* (%)** | | | | |
| One | 25 (32.89) | 21 (38.89) | 19 (44.19) | **0.002**[b] |
| Two | 10 (13.16) | 10 (18.52) | 9 (20.93) | |
| Three | 1 (1.32) | 5 (9.26) | 4 (9.3) | |
| 1 with other HC practitioner | 39 (51.32) | 7 (12.96) | 11 (25.58) | |
| 2 with other HC practitioner | 1 (1.32) | 2 (3.70) | 0 (0) | |

[#]multiple answers possible;

[$]Comparison between Japan and the Philippines only; JP, Japan; PH, the Philippines; TW, Taiwan; ICU, intensive care unit; PT, physical therapy; HC, health care; numbers in bold indicate $p < 0.05$

[a] Pearson chi-square test

[b] Fisher's exact test

[*] One-way ANOVA

incentive spirometry, vibration technique bicycle ergometer exercise, and progressive resistance exercise were used by the respondents from the Philippines significantly more than those from Japan and Taiwan (all $p < 0.05$). Finally, chest expansion technique, percussion technique, active-assisted range of motion, and coughing technique were applied by the respondents from Taiwan more than those from Japan and the Philippines (all $p < 0.05$).

## Challenges in the delivery of PT services in the ICU

The Pearson chi-square test (Fisher's exact test) revealed significant differences in some of the challenges encountered in ICU PT service delivery between the three countries. ($p < 0.05$, Fig 3 and S2 Table).

Compared with the other two countries, the challenge(s) of "no direct access to ICU patients" was encountered significantly more in Japan; "decreased PT ICU exposure," "little-to-no-autonomy in treatment modification," and "provision of PT by ICU nurses instead of physical therapists per se" were encountered more significantly in the Philippines; and others, such as patient-related (e.g., compliance or willingness and case load) treatment-related (e.g., unclear protocol and limited treatment PT interventions to provide or time to treat ICU patients), insurance-related or manpower-related concerns (e.g., indifferences or incoordination between members of the health-care team), were encountered significantly more in Taiwan.

Moreover, the most commonly encountered challenge in all three countries was "little-to-no training prior to ICU duty," with respondents from Japan experiencing it non-significantly more frequently than those from the Philippines and Taiwan ($p = 0.173$).

Due to different sociodemographic profile between three countries, we further assessed whether there was difference in each challenges using logistic regression with adjustment of "age", "highest educational attainment ", and "work experience as physiotherapist". After adjustment, there is no significantly difference between three countries in "No direct access to

**Table 3. ICU-related PT training of respondents.**

| | JP (n = 76) | PH (n = 45) | TW (n = 43) | p value |
|---|---|---|---|---|
| **Participation in ICU-related PT training, *n* (%)** | | | | |
| Yes | 15 (19.74) | 2 (4.44) | 22 (51.16) | **0.000**[a] |
| No | 61 (80.26) | 43 (95.56) | 21 (48.84) | |
| **Training facilitator–Level, *n* (%)** * # | | | | |
| Institutional | 10 (66.67) | 0 (0) | 12 (54.55) | 0.281[b] |
| Local/Regional | 3 (20) | 1 (50) | 8 (36.36) | 0.512[b] |
| National | 12 (80) | 1 (50) | 6 (27.27) | **0.002**[b] |
| International | 1 (6.67) | 0 (0) | 0 (0) | 0.436[b] |
| **Training facilitator–Specialization, *n* (%)** * # | | | | |
| Rehabilitation department | 12 (80) | 2 (100) | 19 (86.36) | 0.765[b] |
| Intensive care unit | 15 (100) | 1 (50) | 3 (13.64) | **0.000**[b] |
| Others | 2 (13.33) | 0 (0) | 3 (13.64) | 1.000[b] |
| **Mode of training, *n* (%)** # | | | | |
| Lecture | 4 (26.67) | 0 (0) | 5 (22.73) | 1.000[b] |
| Demonstration | 1 (6.67) | 0 (0) | 3 (13.64) | |
| Lecture + demonstration | 10 (66.67) | 2 (100) | 14 (63.64) | |
| **Duration of training, *n* (%)** # | | | | |
| 1–4 hours | 2 (13.33) | 1 (50) | 7 (31.82) | 0.348[b] |
| 5–8 hours | 1 (6.67) | 1 (50) | 4 (18.18) | |
| 9–12 hours | 1 (6.67) | 0 (0) | 3 (13.64) | |
| 13–16 hours | 3 (20) | 0 (0) | 2 (9.09) | |
| 17–20 hours | 0 (0) | 0 (0) | 0 (0) | |
| 21–24 hours | 0 (0) | 0 (0) | 0 (0) | |
| >24 hours | 8 (53.33) | 0 (0) | 6 (27.27) | |

*frequency based on those who answered 'yes';

#multiple answers possible; JP, Japan; PH, the Philippines; TW, Taiwan; ICU, intensive care unit; PT, physical therapy; numbers in bold indicate $p < 0.05$

[a] Pearson chi-square test

[b] Fisher's exact test

ICU", "Little-to-no autonomy in treatment modification", and "Provision of PT by ICU nurses instead of PTs perse patients". However, the *p* value of "No direct access to ICU" is near 0.05 ($p = 0.0502$). Moreover, "Little-to-no training prior to ICU duty" was changed from non-significantly to significantly, and "Decreased PT ICU exposure" remains still significantly different between three countries. In "Little-to-no training prior to ICU duty", the odd ratio of Philippines vs Japan and Taiwan vs Japan are 0.154 and 0.376, respectively. These indicate Japanese therapists considered lack of training as a bigger challenge than therapists in the other two countries.

## Discussion

This cross-sectional study explored the extent of PT intervention in the ICU in Japan, the Philippines, and Taiwan. Considerable variations were noted in sociodemographic variables and some ICU-related variables between the three countries.

The World Health Organization has reported marked sex disproportion in the health-care profession [32, 33]. In our study, this was noted in Taiwan, but not in Japan and the Philippines. Because of limited studies on this topic, determining the reason for this disparity

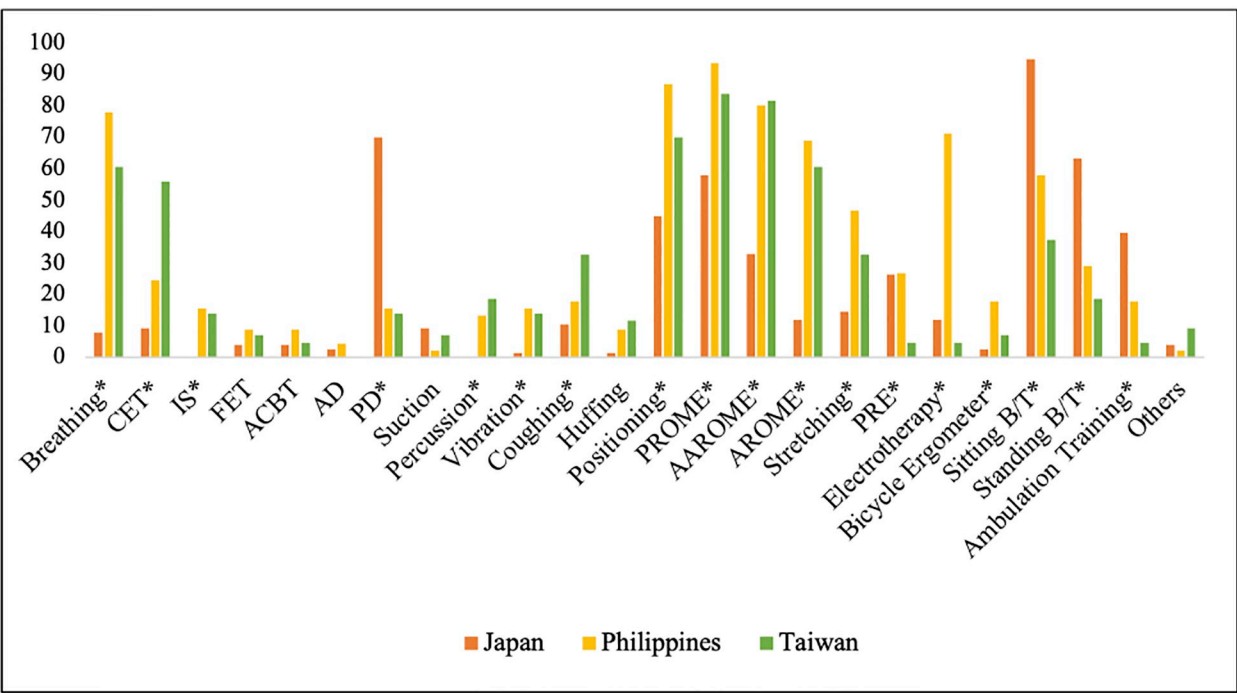

**Fig 2. Between-country comparison of the most commonly implemented PT techniques in the ICU (in %).** $*p < 0.05$, comparison between Japan, the Philippines, and Taiwan; CET, chest expansion technique; IS, incentive spirometry; FET, forced expiration technique; AD, autogenic drainage; PD, postural drainage; PROME, passive range of motion exercise; AAROME, active-assisted range of motion exercise; AROME, active range of motion exercise; PRE, progressive resistance exercise; B/T, balance and tolerance.

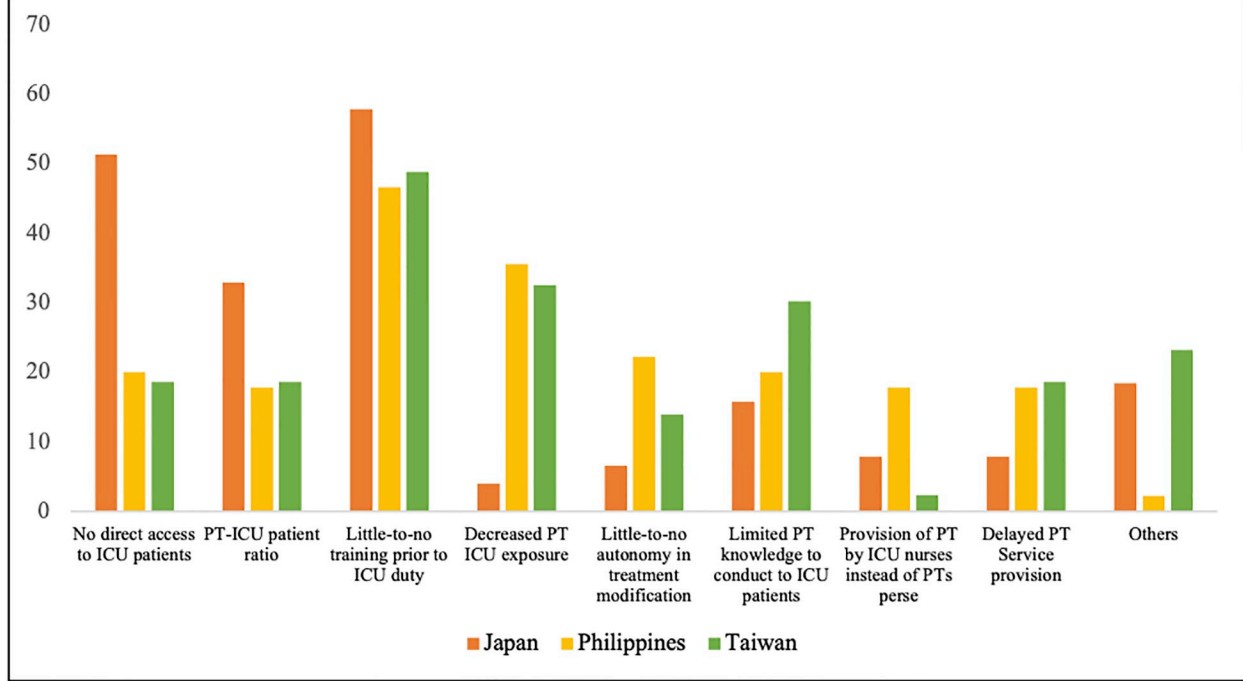

**Fig 3. Challenges in PT implementation in the ICU (in %).** PT, physical therapy; ICU, intensive care unit; PTs, physical therapists.

becomes challenging. On the basis of the World Physiotherapy (WPT) member organization profile, this disparity may be due to the number of practicing physical therapists working in each country. However, the result from our study obtained in the Philippines did not reflect the WPT data but are probably associated with the interventions that are usually implemented in the country, such as positioning technique. Additionally, marked differences in age were noted between the respondents from Japan, the Philippines, and Taiwan. A study indicated that the mean age of ICU physical therapists was 32.42 years [34], which is consistent with that of the respondents in Japan (most of whom were ≥31 years old). However, there were more respondents in the Philippines and Taiwan younger than 30 years, which is similar to the mean age of Indian ICU physical therapists (approximately 20–30 years) [35].

Any relevant PT-related academic degree enables the physical therapist to practice PT, although the details may vary between countries [36]. Our data indicated that the respondents from the Philippines and Taiwan had a bachelor's, master's, or doctorate degree in PT, which was similar to the degrees of Turkish ICU physical therapists [34], whereas the respondents from Japan included diploma holders, which is also similar to previous findings [27, 35]. Notably, we did not assess whether any of the three countries had any entry-level degree specifically for the physical therapist profession in the ICU. The availability of training or institutions offering different degrees in PT vary across countries; this factor may thus contribute to the highest educational attainment of the respondents in this study. Gorgon et al. (2013) noted that a limited number of institutions offered a postgraduate degree in PT in the Philippines [37]; by contrast, Japan [38] and Taiwan [39] offer more postgraduate PT programs. PT work experience can be a discerning criterion for employment. Kim and colleagues (2018) noted a direct correlation between work experience and competency for health-care profession [40]. A study reported that novice physical therapists lack confidence and skills but can improve over time [41]. Most physical therapists working in the ICU have ≥10 years in Jordanian [27] or a mean of 10 years of work experience in Turkey [34], which is comparable to the results obtained in Japan. By contrast, Brito et al. and Masley et al. have reported that most ICU physical therapists have worked for ≤10 years [42, 43], which was in line with the work experience of respondents from the Philippines and Taiwan in the present study. These results, however, did not necessarily reflect the confidence and skills of the respondents; instead, they indicated that these three Asian countries did not require physical therapists to have any work experience to practice in the ICU.

In this study, most respondents from all three countries worked in medical centers or national teaching hospitals. This result may be attributed to the extensive services offered and the existence of more ICUs in these types of hospital [44]. Most ICU physical therapists in Japan and Taiwan reported worked in privately owned hospitals, whereas most ICU physical therapists from the Philippines worked in public hospitals. This may be because of the greater work opportunities in private hospitals in Taiwan, the predominance of private hospitals in Japan [45], and the higher chances of tenure in public hospitals in the Philippines [46]. Physical therapists working in a teaching hospital are required to provide more extensive PT intervention to ICU patients [24]; hence, the high number of respondents working in a teaching hospital depicts the high demand for physical therapists in such a setting.

Physical therapists are an essential part of the critical care team [47]. Managing critically ill patients is challenging; thus, possessing some knowledge for managing these patients is crucial to enhance the quality of PT intervention. According to the standard guideline for ICU PT, experienced physical therapists are required to be present in the ICU (ICS, 2019). However, studies have reported that physical therapists have varying length of ICU work experience—ranging from no experience at all [23, 26, 34] to a mean of 6 [34] or 10 years [23], which is consistent with our results. This may be due to various factors such as brain drain in the

Philippines [48] and the paucity of physical therapists in Taiwan [49]. In Japan, although most respondents were more experienced, this may not be sufficient to meet the demands of the high number of hospitals in the country [50].

Almost all respondents in our study worked in the PT/rehabilitation department. Moreover, most of the respondents in our study do not hold a permanent position in the ICUs, similar to that observed in other countries [51, 52]. Taito et al. explained that physical therapists work on-demand instead of being exclusively in the ICU [51], which may explain the nonexclusive hiring of physical therapists in the ICU in Japan, the Philippines, and Taiwan. As mentioned in recommendations on basic requirements for ICU, the ICU should have an exclusive physical therapist [53] or at least an adequate number of physical therapists [54]. However, only a few countries—typically, European countries—follow these guidelines [55]. Physical therapists play a vital role in responding to the needs of critically ill patients in the ICU by being a part of the multidisciplinary team [56]. Given that full-time PT significantly enhances the recovery of critically ill patients [57], the recommended guidelines should be followed.

In the United States, ICU physical therapists typically conduct PT for neurologic and trauma patients than for medical patients [4]. Conversely, our results revealed that a greater percentage of physical therapists from Japan, the Philippines, and Taiwan worked in the medical ICU than in neurologic or pediatric ICUs. However, the survey was conducted through voluntary participation, which precluded controlling for the number of respondents in a specific specialization or working in the same hospital. Furthermore, the standard guidelines suggest that an adequate number of physical therapists must remain in the ICU 24 h/day every day [54]. However, an evident discrepancy was noted: Most respondents reported ICU work durations of ≤1–2 h a day. Some studies have reported ICU work durations ranging from 3 h (maximum value) [25] to 3.5 h daily (mean) [34]. Patient load, severity of the patient condition, or the exercises provided to the patients may have contributed to such discrepancy. The variation in the daily frequency of treatment per patient can also affect the physical therapists' mean ICU work duration [58].

The standard guidelines also state that on-call physical therapists must be present to respond to the immediate needs of critically ill patients [54]. A study reported that on-call PT services in the ICU were available in private, but not in public, institutions in India [35]. However, this study focused on the general practice of on-call practice in the ICU. Notably, on-call PT service did not exist in Taiwan, whereas some engagement existed in Japan and the Philippines, indicating better compliance with the guidelines. Jones et al. (1992) reported that on-call PT services existed in Australia and the United Kingdom but were limited in Hong Kong. On-call PT in some countries is implemented within the regular work schedule [52, 55], which can be described as being "on-demand" [51], whereas others extend such services through a 24/7 on-call [58], weekend or holiday on-call [23], or on-call during night shifts [23, 35].

Our findings highlight the shortage of ICU physical therapists in Japan, the Philippines, and Taiwan, which is similar to that observed in other studies [27, 34]. The recommendation of one ICU physical therapist to four Level 3 beds [54] was not achieved. In the Philippines, the number physical therapists working in the ICU was twice as high as that in Japan and Taiwan. However, this may just imply that more physical therapists in the Philippines working in the hospital also work in the ICU.

Regarding patient load, studies have reported that an average of 5–10 in Turkey and Brazil [34, 59] to at most 25 ICU patients are treated every day in India [35]. Conversely, most respondents in our study reported managing only 1 or 2 ICU patients a day. This discrepancy may be caused by factors such as knowledge on the importance of PT among ICU patients [27] or referral system/consultation criteria for PT [24, 27]. Physician referral is required for utilizing PT services in some countries [60, 61]. In the present study, Japan, the Philippines,

and Taiwan practice a hierarchical referral system, where referral to PT is provided by either intensivists or physiatrist. Notably, some countries practice a blanket referral system, where ICU patients are automatically referred to PT [60, 62]. Some countries, such as Albania [52] and Sri Lanka [23], practice direct PT access, which can enable the speedy provision of PT to critically ill patients [60] and is cost-effective [63]. Accordingly, a call to action regarding direct access to PT has already been initiated at the international level [64].

PT implementation differs from that of other health-care services. PT is usually delivered one-on-one. Therefore, the standard guideline of assigning one PT to four Level 3 beds [54] can be sometimes deemed unsuitable due to various therapist-related or patient-related factors, such as muscular strength and the presence of tubes and attached equipment in ICU patients. Most physical therapists from the Philippines and Taiwan in our study reported delivering one-on-one PT. Takahashi et al. (2021) mentioned that the early rehabilitation of ICU patients requires a multidisciplinary team [65], which is being practiced in Japan relative to that in the Philippines and Taiwan. Similar to the collaboration between nurses in caring for patients during the COVID-19 pandemic [66], collaboration between health-care workers is equally important for the management of ICU patients.

Availability of ICU-related PT training differs across countries [27, 34], contributing to few trained physical therapists [24, 34, 52]. Similarly, only a few respondents from Japan, the Philippines, and Taiwan in our study reported participating in such training, which does not comply with the international standards necessitating ICU physical therapists to be duly trained to ensure quality PT services [54]. However, the unavailability of such a training precludes compliance with ICU standards. Such training typically also enables the ICU physical therapist to provide appropriate PT interventions according to the patients' needs. In Japan, the Philippines, and Taiwan, the treatment focus was respiratory-related intervention, therapeutic mobilization, or functional training. According to the findings for 50% of respondents who reported having conducted such PT interventions, those from Taiwan frequently implement respiratory-related PT interventions in the ICU, which is similar to that practiced by European [55] and Nepali ICU physical therapists [25]. However, respondents from the Philippines practice therapeutic mobilization exercises more commonly than respiratory-related interventions and functional training, which is similar to the exercises practice of Albanian ICU physical therapists [52]. Nevertheless, studies have demonstrated the equal practice of respiratory-related and therapeutic rehabilitation-related interventions by ICU physical therapists [34, 35]. By contrast, Japanese respondents are more inclined to offer functional training, which is in line with the ICU practice in the United States [43]. Nonetheless, a combination of these PT interventions improve the overall function of critically ill patients [22]. Early progressive mobilization should be implemented in adult ICU patients to obtain favorable patient clinical outcomes [67, 68]. Although the implementation of different PT interventions significantly vary across countries [67], the focus of treatment intervention cannot be based solely on the geographic location. The patients' condition is a more critical factor influencing the decision of the type of PT intervention provided; however, this study failed to investigate this parameter. In Australia, not only do PT interventions vary between states, but also their provision differs between patients based on disease severity [69]. Future studies should expand our research by investigating the differences in specific PT interventions implemented in the ICU based on patient condition in Japan, the Philippines, and Taiwan.

The status of PT service in the ICU may also be contributed by the challenges encountered by physical therapists during the delivery of ICU PT services [24, 27], including limited ICU PT staff [24, 70] and training [52, 71], which was also indicated by our findings. The lack of appropriate PT knowledge before exposure in the ICU and decreased PT ICU exposure are also challenges faced by not only our respondents but also physical therapists in other

countries [24, 27]. Hecimovich and Volet explained the direct correlation between training and professional confidence, which may be a major reason why the lack of training is perceived as challenging [72]. Hence, our findings may reflect the physical therapists' perception of the necessity of such training. Insufficient ICU PT staff is considered a major barrier to PT intervention in the ICU [24, 70], and this shortage exists in Japan, the Philippines, and Taiwan. Future studies should correlate how these challenges affect the quality of PT intervention in the ICU. System-related challenges include the lack of direct access and delay in the provision of ICU PT services. Çakmak et al. (2019) reported that one of the most common barriers in PT intervention in the ICU is the lack of direct PT access [34], which delays the provision of PT services to critically ill patients. Direct PT access is not only cost-effective [63, 73] but also ensures the prompt delivery of ICU PT services [60]. Teamwork-related challenges in Japan, the Philippines, and Taiwan include the lack of professional autonomy for treatment modification, provision of PT by ICU nurses instead of physical therapists, and indifference of team members for the treatment plan. Nonrecognition of professional autonomy among physical therapists was reported to be a barrier in PT intervention in the ICU [34]. The provision of professional autonomy to physical therapists, especially regarding patient treatment planning, may provide better patient outcomes [61] and should therefore be considered in Japan, the Philippines, and Taiwan. Working in an environment that implements good communication within the HC team, achieving common goals with optimism [74], can help better work- and patient-related outcome. Therefore, support should be provided to ICU physical therapists to help them establish a rapport with other health-care professionals. Finally, patient-related challenges, including the lack of compliance with treatment, were noted in this study. Patients should be educated about the advantages of PT for their recovery [34].

This study has several limitations. First, the response rate was low; therefore, the results may not adequately represent all ICU physical therapists in the three countries. Second, no sample size data were available because of the lack of information on the number of ICU physical therapists in each country. Third, the survey mostly comprises closed-ended questions, precluding the respondents from providing rationale for their answers. Moreover, the inability to set limitations on the number of items to choose from might have caused one country to be more involved in a specific area of PT practice than the other. Fourth, there were potential selection bias due to links of surveys in Taiwan and Philippines were post on social media since therapists who use social media may be relatively more proactive in answering online questionnaires or younger. Finally, we did not assess the common assessment, evaluation, and examination methods that are usually conducted for critically ill patients in the ICUs, which are deemed critical to PT practice and may help better delineate the status of PT intervention in the ICU.

## Conclusion

Several characteristics of PT intervention in the ICU vary between Japan, the Philippines, and Taiwan. Although none of the countries seem to impose a strict qualification criterion for physical therapists to work in an ICU, the differences in the ICU-related profile of the respondents between the countries indicate that Japan, the Philippines, and Taiwan comply differently with the internationally recommended PT practice standards. Prioritization in specific PT treatment also differs between Japan, the Philippines, and Taiwan. Finally, regardless of the extent of inclusion of PT in the ICU setting, various challenges to ICU PT service delivery exist in these three countries. Our findings add to the literature regarding the trends and status of PT practice in the ICU. Most importantly, our results underline the need for an open discussion between policymakers and PT organizations in all the three countries regarding the establishment or enhancement of policies and guidelines to match international standards.

## Supporting information

**S1 Table. Most commonly implemented PT techniques in the ICU.**
(PDF)

**S2 Table. Challenges in the delivery of PT services in the ICU.**
(PDF)

**S1 Checklist. STROBE statement checklist.**
(DOCX)

**S2 Checklist. Checklist for Reporting Results of Internet E-Surveys (CHERRIES).**
(DOCX)

**S1 Data.**
(XLSX)

## Acknowledgments

This manuscript was edited by Wallace Academic Editing. We would like to thank Dr. Chia-Ing Li for her valuable help on the logistic regression analysis.

## Author Contributions

**Conceptualization:** Mary Audrey Domingo Viloria, Tetsuya Takahashi, Yu-Jung Cheng.

**Data curation:** Mary Audrey Domingo Viloria, Tetsuya Takahashi, Yu-Jung Cheng.

**Formal analysis:** Mary Audrey Domingo Viloria, Yu-Jung Cheng.

**Funding acquisition:** Shin-Da Lee, Yu-Jung Cheng.

**Methodology:** Mary Audrey Domingo Viloria, Tetsuya Takahashi, Yu-Jung Cheng.

**Supervision:** Yu-Jung Cheng.

**Writing – original draft:** Mary Audrey Domingo Viloria.

**Writing – review & editing:** Mary Audrey Domingo Viloria, Shin-Da Lee, Yu-Jung Cheng.

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
