## [Decision Letter · Decision Letter 0]

23 Jan 2023

PONE-D-22-30062Physical therapy in the intensive care unit: A comparative study of three Asian countriesPLOS ONE

Dear Dr. Cheng,

Thank you for submitting your manuscript to PLOS ONE. After careful consideration, we feel that it has merit but does not fully meet PLOS ONE’s publication criteria as it currently stands. Therefore, we invite you to submit a revised version of the manuscript that addresses the points raised during the review process, which you can find below in this mail.

We look forward to receiving your revised manuscript.

Kind regards,

Elisa Ambrosi

Academic Editor

PLOS ONE

Journal Requirements:

When submitting your revision, we need you to address these additional requirements. 1. Please ensure that your manuscript meets PLOS ONE's style requirements, including those for file naming. The PLOS ONE style templates can be found at https://journals.plos.org/plosone/s/file?id=wjVg/PLOSOne_formatting_sample_main_body.pdf and https://journals.plos.org/plosone/s/file?id=ba62/PLOSOne_formatting_sample_title_authors_affiliations.pdf  2.Please amend your current ethics statement to address the following concerns:a) Did participants provide their written or verbal informed consent to participate in this study?b) If consent was verbal, please explain i) why written consent was not obtained, ii) how you documented participant consent, and iii) whether the ethics committees/IRB approved this consent procedure. 3. Thank you for stating the following financial disclosure:  "This work was supported by the Ministry of Science and Technology in Taiwan (111-2410-H-039-004-), China Medical University Hospital (DMR-110-172), and China Medical University (CMU111-MF-89)". 
Please state what role the funders took in the study.  If the funders had no role, please state: "The funders had no role in study design, data collection and analysis, decision to publish, or preparation of the manuscript." If this statement is not correct you must amend it as needed. Please include this amended Role of Funder statement in your cover letter; we will change the online submission form on your behalf.  4. In your Data Availability statement, you have not specified where the minimal data set underlying the results described in your manuscript can be found. PLOS defines a study's minimal data set as the underlying data used to reach the conclusions drawn in the manuscript and any additional data required to replicate the reported study findings in their entirety. All PLOS journals require that the minimal data set be made fully available. For more information about our data policy, please see http://journals.plos.org/plosone/s/data-availability.Upon re-submitting your revised manuscript, please upload your study’s minimal underlying data set as either Supporting Information files or to a stable, public repository and include the relevant URLs, DOIs, or accession numbers within your revised cover letter. For a list of acceptable repositories, please see http://journals.plos.org/plosone/s/data-availability#loc-recommended-repositories. Any potentially identifying patient information must be fully anonymized.Important: If there are ethical or legal restrictions to sharing your data publicly, please explain these restrictions in detail. Please see our guidelines for more information on what we consider unacceptable restrictions to publicly sharing data: http://journals.plos.org/plosone/s/data-availability#loc-unacceptable-data-access-restrictions. Note that it is not acceptable for the authors to be the sole named individuals responsible for ensuring data access.We will update your Data Availability statement to reflect the information you provide in your cover letter. 

Reviewers' comments:

Reviewer's Responses to Questions

**Comments to the Author**

1. Is the manuscript technically sound, and do the data support the conclusions?

Reviewer #1: Yes

Reviewer #2: Partly

2. Has the statistical analysis been performed appropriately and rigorously? 

Reviewer #1: I Don't Know

Reviewer #2: No

3. Have the authors made all data underlying the findings in their manuscript fully available?

Reviewer #1: Yes

Reviewer #2: Yes

4. Is the manuscript presented in an intelligible fashion and written in standard English?

Reviewer #1: Yes

Reviewer #2: Yes

5. Review Comments to the Author

Reviewer #1: Dear Authors

Thanks a lot for the opportunity you have offered me to revise the fascinating manuscript " Physical therapy in the intensive care unit: A comparative study of three Asian countries". I thank the authors for their effort in producing this exciting review during the COVID-19 pandemic. It is perfectly aligned with my area of research and expertise; thus, I am confident to offer a valuable peer review.

As a significant strength, this manuscript describe the practice of PT in the intensive care unit (ICU), which differs between various country. This proposal is interesting in the field and adds information to the existing evidence in the literature.

As a major weakness, the manuscript sometimes lacks details and clarity concerning methodological steps that would help improve the understanding of the manuscript. Therefore, I have suggested some strategies to improve authors' reporting and increase the quality of their work.

Overall, my peer-review is a major revision: I suggest revising the manuscript to improve the pitfalls presented. The final goal is to improve the overall clarity of the message to help the reader understand this fundamental topic. I look forward to reading the revised version of the manuscript.

Thanks again, and good luck with researching in this challenging time.

@Major revisions

#METHODS

*General. I suggest that authors adhere to the suggested reporting for STROBE observational studies (https://www.thelancet.com/journals/lancet/article/PIIS0140-6736(07)61602-X/fulltext#article_upsell). Furthermore, if their survey is electronic, I recommend that they also use the reporting of CHERRIES (https://www.ncbi.nlm.nih.gov/pmc/articles/PMC1550605/). Accordingly, authors should reorganize the manuscript following these guidelines for the sections materials and methods, results and discussion. Moreover, they should declare clearly the adoption of this guideline in the Methods section.

*Survey development. The authors should be clearer. How was the survey created (e.g., which references and how did it differ from what existed in the literature)? How many questions made up each section? What type of questions were used (e.g. open or closed)? How many answer possibilities were there (e.g. single or multiple)? How many pages were there? These elements should be presented in more detail. Furthermore, the authors should include the survey in the original target languages and in English as Supplementary files.

*Survey testing. Was the survey tested with a pilot? Please be specific.

*Administration. How was the survey administered (paper or online)? How were the participants contacted? Was it free to fill out, or was there an obligation? How long did it last? Were there any economic incentives for completion? How was the possibility of multiple compilations avoided?

*Analysis of the survey. How were missing data handled?

@Minor revisions

#TITLE:

*I suggest authors add in the title the study design (e.g., : an observational study)

#ABSTRACT

*result. Please add statistical values to sustain your statement (e.g., %, n= …).

#KEYWORDS:

*I suggest authors add "rehabilitation", "physical medicine" "Physiotherapists", "survey".

#RESULTS:

*The rate of response is low. This point is a limitation of authors' study. I suggest they declare it in the limitation section.

#DISCUSSION:

*General. The discussion is well organized, but it is a little bit longer. I suggest authors shorten and focus more on it. For example, it is better to avoid the repetition of the results. Discussing the findings with existing evidence or comparing them with different scenarios (e.g., European and American countries) is suitable.

*Direct access of PT. The topic of direct access in physiotherapy is very relevant and strongly discussed also in other fields (e.g. musculoskeletal, e.g. Maselli et al. Direct Access to Physical Therapy: Should Italy Move Forward? Int J Environ Res Public Health. 2022 Jan 4;19(1):555. doi: 10.3390/ijerph19010555). I think this point needs to be developed by emphasising that there is a call to action at international level to develop direct access patient management skills.

*Interprofessional work. Authors have reported the importance of interprofessional work (e.g., between physiotherapists and nurses) in the ICU setting. This element emerged strongly also from different qualitative studies involving nurses, especially during the COVID-19 pandemic. For example, Fontanini et al. have reported the importance of interprofessional work in the ICU setting. I suggest authors look at this paper for their discussion (Italian Nurses' experiences during the COVID-19 pandemic: a qualitative analysis of internet posts. Int Nurs Rev. 2021 Jun;68(2):238-247. doi: 10.1111/inr.12669).

*Managerial perspectives. Authors frequently suggest practical actions for managers. Some of these actions are similar to those proposed for emotional intelligence leaders in managing complex healthcare scenarios (e.g., improve communication, awareness, multidisciplinary works, etc). I suggest authors look at this recent publication in the rehabilitation field by Dell'Isola's research group. Maybe, this publication can add some points to the authors' discussion. (COVID-19 and Health Care Leaders: How Could Emotional Intelligence Be a Helpful Resource During a Pandemic? Phys Ther. 2021 Sep 1;101(9):pzab143. doi: 10.1093/ptj/pzab143).

*Limitations. The rate of response is low. This point is a limitation of the authors' study. I suggest they declare it in the limitation section. Moreover, in this section, the authors should call for future research in the field aimed at overcoming the limits presented in their study.

#CONCLUSION

*Implications. According to their findings, I suggest authors organize this section in implications for clinical practice, education and management. This will be very informative.

#LANGUAGE

*There is a need for a language revision by a native English speaker. For example, there are different grammar mistakes. Moreover, various too many long sentences are presented, limiting the manuscript's readability. Please have a look at the language of the manuscript.

Reviewer #2: Dear Author,

First of all, why did the author choose few Asian countries and the major flaw is validity of the questionnaire used. The questionnaire was not translated as per the standard protocol.

The sample size is small.

The justification to perform the study is not provided

6. PLOS authors have the option to publish the peer review history of their article (what does this mean?). If published, this will include your full peer review and any attached files.

Reviewer #1: No

Reviewer #2: No

---

## [Author Response · Author response to Decision Letter 0]

15 Mar 2023

Reviewer #1:

Major revisions

1. I suggest that authors adhere to the suggested reporting for STROBE observational studies. (https://www.thelancet.com/journals/lancet/article/PIIS0140-6736(07)61602-X/fulltext#article_upsell). 

Response: Thanks for reviewer’s suggestion. We have added STROBE Statement – Checklist as a supplementary document.

2. If their survey is electronic, I recommend that they also use the reporting of CHERRIES (https://www.ncbi.nlm.nih.gov/pmc/articles/PMC1550605/). 

Response: Thanks for reviewer’s suggestion. We have used reporting of CHERRIES to check our manuscript. However, not all items were mentioned in manuscript.

3. Authors should reorganize the manuscript following these guidelines for the sections materials and methods, results and discussion. 

Response: Thanks for reviewer’s suggestion. We have re-reorganize the manuscript following guidelines.

3.1 They should declare clearly the adoption of this guideline in the Methods section.

Response: Thanks for reviewer’s suggestion. The survey was based on the study of Sigera et al. on 2016. We removed questions of ICU research and on-call service. In addition, we modified/included several questions such as the type of hospital, work experience, most common physiotherapy interventions, and common challenge experienced in the ICU. 

3.2 How many questions made up each section?

Response: Thanks for reviewer’s suggestion. There are two sections and total 21 questions. 

- Section I, 7. 

- Section II, 14. 

3.3. What type of questions were used?

Response: Type of questions used identified. (line 119)

3.4. How many answer possibilities were there?

Response: The answer possibilities are different in each question. All are multiple answeres. 

3.5. How many pages were there?

Response: There were a total of eight (8) sections in the google form including the informed consent section and instruction section. 

3.6. Survey testing. Was the survey tested with a pilot? Please be specific.

Response: No pilot-testing of the survey made and hence included as a limitation.. However, the e-survey of Taiwanese version was validated by two PTs who currently work in ICU. After they gave suggestion, some questions were edited. 

3.6. Administration. How was the survey administered (paper or online)? How were the participants contacted? Was it free to fill out, or was there an obligation? How long did it last? Were there any economic incentives for completion? How was the possibility of multiple compilations avoided?

Response: The survey is e-survey. Responses were collected via Google Forms. The link of electronic surveys was posted on social media of PT interest groups in Taiwan and Philippine, and Japanese Society of Education for Physicians and Trainees in Intensive Care (JSEPTIC) in Japan. The Google Forms were opened from November 2021 to February 2022. There was no economic incentives and the e-survey is free to fill out. Once the answer was sent, there was no possibility for modification. 

3.7. Analysis of the survey. How were missing data handled?

Response: All participants completed the survey. No missing data was needed for handling. 

Minor revisions

4. TITLE: I suggest authors add in the title the study design (e.g., : an observational study)

Response: Thanks for reviewer’s suggestion. We have changed title to “Physical therapy in the intensive care unit: An observational comparative study of three Asian countries” 

5. ABSTRACT: result. Please add statistical values to sustain your statement (e.g., %, n= …).

Response: Thanks for reviewer’s suggestion. Abstract was revised. 

5. KEYWORDS: I suggest authors add "rehabilitation", "physical medicine" "Physiotherapists", "survey".

Response: Keywords were added 

6. RESULTS: The rate of response is low. This point is a limitation of authors' study. I suggest they declare it in the limitation section.

Response: Low response rate is included in the limitation. 

7. DISCUSSION: General. The discussion is well organized, but it is a little bit longer. I suggest authors shorten and focus more on it. For example, it is better to avoid the repetition of the results. Discussing the findings with existing evidence or comparing them with different scenarios (e.g., European and American countries) is suitable.

Response: 

- As the STROBE checklist was suggested to be used, a summary of the study was included in the first part of the discussion. Nevertheless, the longer version of the summarized part was cut short to only two sentences. 

- Sociodemographic profile was cut a little shorter. 

- The repetition in discussion was removed. 

8. Direct access of PT. The topic of direct access in physiotherapy is very relevant and strongly discussed also in other fields (e.g. musculoskeletal, e.g. Maselli et al. Direct Access to Physical Therapy: Should Italy Move Forward? Int J Environ Res Public Health. 2022 Jan 4;19(1):555. doi: 10.3390/ijerph19010555). I think this point needs to be developed by emphasising that there is a call to action at international level to develop direct access patient management skills.

Response: 

- Due to the medical regulations in JP, TW, and PH, there is no direct access except certain conditions such as health promotion and sports injury prevention. 

- Suggestion to emphasize that a call to action regarding direct access was included. (line 380) 

9. Interprofessional work. Authors have reported the importance of interprofessional work (e.g., between physiotherapists and nurses) in the ICU setting. This element emerged strongly also from different qualitative studies involving nurses, especially during the COVID-19 pandemic. For example, Fontanini et al. have reported the importance of interprofessional work in the ICU setting. I suggest authors look at this paper for their discussion (Italian Nurses' experiences during the COVID-19 pandemic: a qualitative analysis of internet posts. Int Nurs Rev. 2021 Jun;68(2):238-247. doi: 10.1111/inr.12669).

Response: Thanks for reviewer’s suggestion. We have added the importance of interprofessional work in the ICU. 

10. Managerial perspectives. Authors frequently suggest practical actions for managers. Some of these actions are similar to those proposed for emotional intelligence leaders in managing complex healthcare scenarios (e.g., improve communication, awareness, multidisciplinary works, etc). I suggest authors look at this recent publication in the rehabilitation field by Dell'Isola's research group. Maybe, this publication can add some points to the authors' discussion. (COVID-19 and Health Care Leaders: How Could Emotional Intelligence Be a Helpful Resource During a Pandemic? Phys Ther. 2021 Sep 1;101(9):pzab143. doi: 10.1093/ptj/pzab143).

Response: Thanks for reviewer’s suggestion. Included how good relationship among members of the HC team could help improve work and patient-related outcome. (line 390)

11. Limitations. The rate of response is low. This point is a limitation of the authors' study. I suggest they declare it in the limitation section. Moreover, in this section, the authors should call for future research in the field aimed at overcoming the limits presented in their study.

Response: We have Included the “low response rate” as limitation. 

12. CONCLUSION: Implications. According to their findings, I suggest authors organize this section in implications for clinical practice, education and management. This will be very informative.

Response: Thanks for reviewer’s suggestion. Included clinical implications. (lines 413-416)

13. LANGUAGE: There is a need for a language revision by a native English speaker. For example, there are different grammar mistakes. Moreover, various too many long sentences are presented, limiting the manuscript's readability. Please have a look at the language of the manuscript.

Response: The revised manuscript has been edited by professional English editors. 

Reviewer #2: 

Dear Author,

1. First of all, why did the author choose few Asian countries and the major flaw is validity of the questionnaire used. The questionnaire was not translated as per the standard protocol.

Response: Thanks for reviewer’s suggestion. We chose these three Asia countries due to Japan, Philippines, and Taiwan have different healthcare system, and there is no study regarding the current status of PT practice in the ICU. About translation, yes, questionnaires was no translated according to standard protocol. However, Professor Takahashi, also as a PT, is the Executive Director of The Japanese Society of Intensive Care Medicine. We believe his participation is sufficient to increase the validity of the questionnaire.

2. The sample size is small.

Response: Thanks for reviewer’s suggestion. Included small sample size seen as low response rate which is included in the limitation. 

3. The justification to perform the study is not provided

Response: Although numerous studies have reported on the current status of PT intervention in the ICU, no study has compared it between different Asian countries. Since Japan, Philippines, and Taiwan have different health-care systems, we compared the extent of PT intervention in the ICU between these three countries.

---

## [Decision Letter · Decision Letter 1]

26 Apr 2023

PONE-D-22-30062R1Physical therapy in the intensive care unit: An observational comparative study of three Asian countriesPLOS ONE

Dear Dr. Cheng,

Thank you for submitting your manuscript to PLOS ONE. After careful consideration, we feel that it has merit but does not fully meet PLOS ONE’s publication criteria as it currently stands. Therefore, we invite you to submit a revised version of the manuscript that addresses the points raised during the review process.

We look forward to receiving your revised manuscript.

Kind regards,

Elisa Ambrosi

Academic Editor

PLOS ONE

Journal Requirements:

Reviewers' comments:

Reviewer's Responses to Questions

**Comments to the Author**

1. If the authors have adequately addressed your comments raised in a previous round of review and you feel that this manuscript is now acceptable for publication, you may indicate that here to bypass the “Comments to the Author” section, enter your conflict of interest statement in the “Confidential to Editor” section, and submit your "Accept" recommendation.

Reviewer #1: All comments have been addressed

Reviewer #3: (No Response)

2. Is the manuscript technically sound, and do the data support the conclusions?

Reviewer #1: Yes

Reviewer #3: Partly

3. Has the statistical analysis been performed appropriately and rigorously? 

Reviewer #1: I Don't Know

Reviewer #3: No

4. Have the authors made all data underlying the findings in their manuscript fully available?

Reviewer #1: Yes

Reviewer #3: Yes

5. Is the manuscript presented in an intelligible fashion and written in standard English?

Reviewer #1: Yes

Reviewer #3: Yes

6. Review Comments to the Author

Reviewer #1: Dear Authors

Thanks a lot for the opportunity you have offered me to revise again the fascinating manuscript " Physical therapy in the intensive care unit: A comparative study of three Asian countries". I thank the authors for having improved the manuscript following the suggestions of reviewers.

Despite the ameliorations, I invite the authors to add the reference of the STROBE and CHERRIES in the main manuscript.

Thus, after having resolved these minor revisions, the paper will be ready to be accepted and published.

I am available to revise the paper again.

Best regards.

Reviewer #3: Dear author, here are my suggestions:

1. I suggest replacing the term “observational comparative” with “cross-sectional study” in all the manuscript

2. Please, insert in the method section all the reporting guidelines that you have used to write the manuscript (e.g. “This protocol is reported in accordance with …)

3. In the method section, please justify how you decided to use Fisher’s exact test instead of the chi–square test. Also, specify how you evaluated all the statistical test assumptions.

4. The study suffers from a possible selection bias. Indeed, physical therapists (PT) who use social media do not represent all the PT who work in an ICU setting (e.g., probably younger and more prone to use online resources for their training). Therefore, I suggest inserting this limitation of the study in the discussion.

5. The statistical analysis conducted (univariable hypothesis testing) does not permit to conclude that the interventions applied in the ICU setting are different because of country differences. The sample of PT in the three countries analyzed is different for significant variables such as age and PT-patient ratio. Therefore, there are many confounders in the relationship of PT intervention – country. I suggest conducting a regression analysis adjusting for possible cofounders to determine the predictors of the different interventions applied. Otherwise (e.g., the sample size is too small to perform a regression), I suggest inserting the lack of an adjustment in the analysis as a major limitation of the study.

Kind regards

7. PLOS authors have the option to publish the peer review history of their article (what does this mean?). If published, this will include your full peer review and any attached files.

Reviewer #1: No

Reviewer #3: No

---

## [Author Response · Author response to Decision Letter 1]

23 May 2023

Reviewers' comments:

Reviewer #1: Dear Authors

Thanks a lot for the opportunity you have offered me to revise again the fascinating manuscript " Physical therapy in the intensive care unit: A comparative study of three Asian countries". I thank the authors for having improved the manuscript following the suggestions of reviewers.

Despite the ameliorations, I invite the authors to add the reference of the STROBE and CHERRIES in the main manuscript.

Thus, after having resolved these minor revisions, the paper will be ready to be accepted and published.

I am available to revise the paper again.

Best regards.

Response: Thanks for reviewer’s suggestion. We have added the references of STROBE and CHERRIES in manuscript.

Revised: 

Line 109:

We used a two-part, semistructured, open online survey which based on Equator Network's Checklist for Reporting Results of Internet E-Surveys (CHERRIES) guidelines [29].

Line 125: 

The final survey was evaluated with the Strengthening the Reporting of Observational Studies in Epidemiology (STROBE) statement [30] (S3).

Reviewer #3: Dear author, here are my suggestions:

1. I suggest replacing the term “observational comparative” with “cross-sectional study” in all the manuscript

Response: Thanks for reviewer’s suggestion. We have changed all “observational comparative” with “cross-sectional study” in manuscript. 

2. Please, insert in the method section all the reporting guidelines that you have used to write the manuscript (e.g. “This protocol is reported in accordance with …)

Response: Thanks for reviewer’s suggestion. We have added the references of CHERRIES in manuscript.

Revised: 

Line 109

We used a two-part, semistructured, open online survey which based on Equator Network's Checklist for Reporting Results of Internet E-Surveys (CHERRIES) guidelines [29].

3. In the method section, please justify how you decided to use Fisher’s exact test instead of the chi–square test. Also, specify how you evaluated all the statistical test assumptions.

Response: Thanks for reviewer’s suggestion. We have added the description of when to use Fisher’s exact test at “Statistical Analyses” section. 

Revised: 

Line 127

The data were analyzed using IBM SPSS Statistics for Windows, Version 22.0. (Armonk, NY: IBM Corp) and SAS 9.4 (SAS Institute Inc., Cary, NC, USA). The respondents’ sociodemographic and ICU-related characteristics are presented as frequency (f), percentage (%), and mean. For the categorical variables, chi-square or Fisher’s exact test were used, and Fisher’s exact test was conducted instead of chi-square test if any of the expected values in a cell is less than five. The one-way analysis of variance (ANOVA) was used for continuous variables. The level of significance was set at p < 0.05. Although the sample size was not determined, sample size between 30 and 500 for comparative analysis was considered appropriate [31].

4. The study suffers from a possible selection bias. Indeed, physical therapists (PT) who use social media do not represent all the PT who work in an ICU setting (e.g., probably younger and more prone to use online resources for their training). Therefore, I suggest inserting this limitation of the study in the discussion. 

Response: Thanks for reviewer’s suggestion. We have added this limitation at “Discussion” section. 

Revised: 

Line 416

Fourth, there were potential selection bias due to links of surveys in Taiwan and Philippines were post on social media since therapists who use social media may be relatively more proactive in answering online questionnaires or younger. 

5. The statistical analysis conducted (univariable hypothesis testing) does not permit to conclude that the interventions applied in the ICU setting are different because of country differences. The sample of PT in the three countries analyzed is different for significant variables such as age and PT-patient ratio. Therefore, there are many confounders in the relationship of PT intervention – country. I suggest conducting a regression analysis adjusting for possible cofounders to determine the predictors of the different interventions applied. Otherwise (e.g., the sample size is too small to perform a regression), I suggest inserting the lack of an adjustment in the analysis as a major limitation of the study.

Response: Thanks for reviewer’s suggestion. We have assessed data of “challenges” using logistic regression with adjustment of “age”, “highest educational attainment “, and “work experience as physiotherapist”. Indeed, after adjustment, there are several data became non-significant different after logistic regression. Thus, we added one paragraph in result section. 

Revised:

Line 225:

Due to different sociodemographic profile between three countries, we further assessed whether there was difference in each challenges using logistic regression with adjustment of “age”, “highest educational attainment “, and “work experience as physiotherapist”. After adjustment, there is no significantly difference between three countries in “No direct access to ICU”, “Little-to-no autonomy in treatment modification”, and “Provision of PT by ICU nurses instead of PTs perse patients”. However, the p value of “No direct access to ICU” is near 0.05 (p = 0.0502). Moreover, “Little-to-no training prior to ICU duty” was changed from non-significantly to significantly, and “Decreased PT ICU exposure” remains still significantly different between three countries. In “Little-to-no training prior to ICU duty”, the odd ratio of Philippines vs Japan and Taiwan vs Japan are 0.154 and 0.376, respectively. These indicate Japanese therapists considered lack of training as a bigger challenge than therapists in the other two countries.

---

## [Decision Letter · Decision Letter 2]

28 Jul 2023

Physical therapy in the intensive care unit: A cross-sectional study of three Asian countries

PONE-D-22-30062R2

Dear Dr. Cheng,

We’re pleased to inform you that your manuscript has been judged scientifically suitable for publication and will be formally accepted for publication once it meets all outstanding technical requirements.

Kind regards,

Elisa Ambrosi

Academic Editor

PLOS ONE

Additional Editor Comments (optional):

Reviewers' comments:

Reviewer's Responses to Questions

**Comments to the Author**

1. If the authors have adequately addressed your comments raised in a previous round of review and you feel that this manuscript is now acceptable for publication, you may indicate that here to bypass the “Comments to the Author” section, enter your conflict of interest statement in the “Confidential to Editor” section, and submit your "Accept" recommendation.

Reviewer #1: All comments have been addressed

Reviewer #3: All comments have been addressed

2. Is the manuscript technically sound, and do the data support the conclusions?

Reviewer #1: Yes

Reviewer #3: Yes

3. Has the statistical analysis been performed appropriately and rigorously? 

Reviewer #1: Yes

Reviewer #3: Yes

4. Have the authors made all data underlying the findings in their manuscript fully available?

Reviewer #1: Yes

Reviewer #3: Yes

5. Is the manuscript presented in an intelligible fashion and written in standard English?

Reviewer #1: Yes

Reviewer #3: Yes

6. Review Comments to the Author

Reviewer #1: Dear Authors

Thanks a lot for the opportunity you have offered me to revise again the fascinating manuscript " Physical therapy in the intensive care unit: A comparative study of three Asian countries". I thank the authors for having improved the manuscript following the all my suggestions. Thus, the paper is ready to be accepted and published. Congratulations. Best regards.

Reviewer #3: The authors have adequately addressed all my comments raised in a previous round of review. My reccomandation is to 'Accept' the manuscript.

7. PLOS authors have the option to publish the peer review history of their article (what does this mean?). If published, this will include your full peer review and any attached files.

Reviewer #1: No

Reviewer #3: **Yes: **Daniel Feller

---

## [Editor Report · Acceptance letter]

31 Oct 2023

PONE-D-22-30062R2 

Physical therapy in the intensive care unit: A cross-sectional study of three Asian countries 

Dear Dr. Cheng:

I'm pleased to inform you that your manuscript has been deemed suitable for publication in PLOS ONE. Congratulations! Your manuscript is now with our production department. 

Kind regards, 

on behalf of

Dr. Elisa Ambrosi 

Academic Editor

PLOS ONE